# High-Precision Automatic Identification of Fentanyl-Related Drugs by Terahertz Spectroscopy with Molecular Dynamics Simulation and Spectral Similarity Mapping

**DOI:** 10.3390/ijms231810321

**Published:** 2022-09-07

**Authors:** Fangfang Qu, Lei Lin, Pengcheng Nie, Zhengyan Xia

**Affiliations:** 1College of Biosystems Engineering and Food Science, Zhejiang University, Hangzhou 310058, China; 2College of Mechanical and Electrical Engineering, Fujian Agriculture and Forestry University, Fuzhou 310002, China; 3Key Laboratory of Spectroscopy Sensing, Ministry of Agriculture and Rural Affairs, Hangzhou 310058, China; 4School of Medicine, Zhejiang University City College, Hangzhou 310015, China

**Keywords:** terahertz spectroscopy, fentanyl-related drugs, spectral similarity mapping, density functional theory, molecular dynamics simulation

## Abstract

Fentanyl is a potent opioid analgesic with high bioavailability. It is the leading cause of drug addiction and overdose death. To better control the abuse of fentanyl and its derivatives, it is crucial to develop rapid and sensitive detection methods. However, fentanyl-related substrates undergo similar molecular structures resulting in similar properties, which are difficult to be identified by conventional spectroscopic methods. In this work, a method for the automatic identification of 8 fentanyl-related substances with similar spectral characteristics was developed using terahertz (THz) spectroscopy coupled with density functional theory (DFT) and spectral similarity mapping (SSM). To characterize the THz fingerprints of these fentanyl-related samples more accurately, the method of baseline estimation and denoising with sparsity was performed before revealing the unique molecular dynamics of each substance by DFT. The SSM method was proposed to identify these fentanyl analogs based on weighted spectral cosine–cross similarity and fingerprint discrete Fréchet distance, generating a matching list by stepwise searching the entire spectral database. The top matched list returned the identification results of the target fentanyl analogs with accuracies of 94.48~99.33%. Results from this work provide algorithms’ increased reliability, which serves as an artificial intelligence-based tool for high-precision fentanyl analysis in real-world samples.

## 1. Introduction

Fentanyl (N-phenyl-N-[l-(2-phenylethyl)-4-piperidinyl]propanamide) is the prototype of a 4-anilinopiperidine class of synthetic opioids that was first synthesized by Paul Janssen in 1959 [1,2]. As a high lipophilic affinity µ-opioid receptor agonist that rapidly crosses through the cell membranes, fentanyl presents approximately 100 times more potency than the highly ionized and hydrophilic morphine based on its success as a prescription over the last decades [3,4]. Its pharmacological effects like analgesia and anesthesia could be retained or enhanced by modifying the functional group on fentanyl [5,6]. This makes it easy to synthesize fentanyl analogs by replacing the phenethyl group with various other groups (except hydrogen); displacing the benzene ring attached to the nitrogen atom with substituted or unsubstituted monocyclic aromatic groups; substituting the propionyl group in the 4-anilido fragment with other acyl groups; and introducing substituents (e.g., methyl groups and methoxycarbonyl groups) at different positions on the piperidine ring [7,8]. According to the 25-year (1995–2020) monitoring data reported by the European Monitoring Centre for Drugs and Drug Addiction (EMCDDA), 36 new synthetic fentanyl derivatives have been detected in the European drug market since 2009 [9]. Recently, new fentanyl and its analogs, such as methoxyacetylfentanyl, acetylfentanyl, furanylfentanyl, butyrylfentanyl, 4-fluoroisobutyrfentanyl, carfentanil, and valerylfentanyl, have emerged in the illicit market as cheaper and more potent alternatives to heroin [10]. Acute intoxications and related fatalities caused by the abuse of fentanyl analogs have become a concerning global health threat [11,12]. Although many relevant programs and regulations have been launched nationwide to control the abuse of fentanyl-related substances, a series of non-pharmaceutical fentanyl analogs have been derived as illegal drugs that escape law enforcement from the local government [13,14].

The fundamental strategy to address this crisis is to develop fentanyl detection technology. To date, the most commonly used method to identify fentanyl and its analogs are chromatography–mass spectrometry, such as gas chromatography–mass spectrometry [15,16], liquid chromatography–mass spectrometry [17,18], and liquid chromatography–tandem mass spectrometry [19,20]. Although these methods are accurate and sensitive, they cannot meet the increasing requirements for rapid detection (e.g., in situ and street detection) due to their time-consuming processing procedures conducted in the laboratory [21]. In recent years, studies on rapid detection methods based on immunoassays, including fluorescent immunoassay [22], homogeneous enzyme immunoassay [23], enzyme-linked immunosorbent assay [24], electrochemical immunosensor [25], and lateral flow immunochromatography [26], have also been reported. These studies have greatly enriched research on fentanyl analogs, especially for their metabonomics but challenges remain in meeting the higher requirements for sensitivity. One promising method of identifying fentanyl-related substances is through terahertz (THz) fingerprint spectroscopy because this technique can probe for both the inter-and intra-molecular vibrational modes of crystalline materials (fentanyl analogs are primarily powdered crystalline substances), yielding unique, molecularly-specific vibrational spectra and thus enabling unique identity to chemical species. It should be noted that the fentanyl-related molecules undergo similar structural and metabolic pathways, resulting in similar properties and metabolite production to them. Thus, it is difficult to identify the unique features of each fentanyl-related substance via conventional spectroscopic methods, such as near-infrared spectroscopy and ultraviolet spectrophotometry. THz time-domain spectroscopy (THz-TDS), as a rapid and sensitive technique to study the dynamics of molecular morphological structure, provides both chemical and structural information for fentanyl analysis. It enables THz-TDS to distinguish between fentanyl-related substances with relatively similar chemical structures, which facilitates the development of specific THz spectral databases of fentanyl and its analogs [27]. We believe that the intrinsic THz fingerprints of these fentanyl analogs can be used not only to improve the knowledge of their basic material properties but also for identification purposes.

This paper presents a method to calculate spectral matching accuracy, aiming to identify fentanyl analogs from the THz spectral database rapidly, automatically, and efficiently. Herein, we collected the THz absorbance spectra of 8 fentanyl-related substances by measuring pressed-pellet samples with a THz-TDS spectrometer. To characterize the THz fingerprints of these fentanyl-related samples more accurately, the method of baseline estimation and denoising with sparsity (BEADS) was performed before revealing the unique molecular dynamics of each substance by density functional theory (DFT). Specifically, we proposed an algorithm named spectral similarity mapping (SSM), a procedure that automatically calculates THz absorbance spectral similarity based on weighted spectral cosine–cross similarity (CCS) and fingerprint discrete Fréchet distance (DFD). It generates a matching list of entries with similar spectra to target analytes by searching the spectral database. Consequently, the top matching result returns quantifiable certainty to infer the identity of the target analyte. This spectral similarity matching and mapping strategy may even realize the automatic identification of counterfeit or illicit fentanyl drugs according to the THz spectral characteristics of the main components of fentanyl.

## 2. Results and Discussion

### 2.1. Molecular Geometric Configuration

The molecular geometric configuration of the fentanyl-related substances, referring to fentanyl (C_22_H_28_N_2_O, CAS: 437-38-7, MS: 336.47), methoxyacetylfentanyl (C_22_H_28_N_2_O_2_, CAS: 101345-67-9, MS: 352.47), acetylfentanyl (C_21_H_26_N_2_O, CAS:3258-84-2, MS: 322.44), furanylfentanyl (C_24_H_26_N_2_O_2_, CAS: 101345-66-8, MS: 374.48), butyrylfentanyl (C_23_H_30_N_2_O, CAS: 1169-70-6, MS: 350.50), 4-fluoroisobutyrfentanyl (C_23_H_29_FN_2_O, CAS: 244195-32-2, MS: 368.49), carfentanil (C_24_H_30_N_2_O_3_, CAS: 59708-52-0, MS: 394.51), and valerylfentany (C_24_H_32_N_2_O, CAS: 122882-90-0, MS: 364.52), were tightly optimized using B3LYP/6-311G in Gaussian 2016 software (Gaussian Inc., Wallingford, CT, USA). Figure 1 shows the molecular electrostatic potential (MEP) of the optimized molecular structures. It can be seen from Figure 1 that all these eight molecules have similar structures to piperidine rings, as well as the structures of phenyl group directly linked to two nitrogen atoms (in blue) by a monocyclic aromatic group. Furthermore, the maximum MEP value of these eight fentanyl substances appeared around the oxygen atoms (in red). These results indicate that similar fingerprint characteristics may arise at THz frequency due to similar molecular vibrational modes of these fentanyl-related substances.

### 2.2. Comparison of Experimental and Theoretical Spectra

According to the optimized molecular geometry of fentanyl-related substances, their theoretical spectra were simulated by DFT. To improve the spectral resolution, the BEADS method was used to reduce the noise and correct the baseline before comparing the experimentally obtained THz absorption spectra with the DFT simulated spectra. As shown in Figure 2, THz absorption spectra and peaks at different frequencies (marked with red dots) were found in the experimental spectra of fentanyl-related substances. The peaks with high intensity were successfully marked by an automatic peak-finding algorithm. However, several spectral characteristics marked with green circles, e.g., 2.83 THz and 3.27 THz of fentanyl (Figure 2b), 1.07 THz of both acetylfentanyl (Figure 2d) and butyrylfentanyl (Figure 2f), and 0.91 THz of valerylfentanyl (Figure 2i), were difficult to be determined whether these are the fingerprint peaks or not due to low resolution. Spectral processing results show that the spectral baselines were accurately estimated (dash-dotted curves) and these indistinct peaks were highlighted in the BEADS-processed THz spectra (black curves). The baseline-corrected experimental spectra and DFT simulated theoretical spectra (blue curves) were in high agreement except with slight frequency shifts and several absorption peaks missing (marked with red circles), which was conducive to improving theoretical analysis accuracy of the formation mechanism of THz absorption peaks. The discrepancy between the experimental and theoretical spectra was mainly due to the different states of the tested sample. Because the experimental samples were compressed pellets of solid powders, the DFT simulations were conducted based on the isolated molecules. Therefore, the intermolecular interaction, crystal field effect, and crystal resonance were excluded in theoretical simulations. Furthermore, the experimental measurements were carried out at 294 K, but the simulations were conducted based on 0 K, thus ignoring the thermal effect. There were more theoretical absorption peaks than the experimental ones, which might be due to the limitation of THz experimental instruments.

### 2.3. Assignment of Absorption Peaks

The formation mechanism of the THz characteristic absorption peaks was analyzed using the visualization function in GaussView 5.08 software (Gaussian Inc., Wallingford, CT, USA). Table 1 lists the assignment of the molecular vibration modes according to the DFT simulated results. There were six peaks for fentanyl, in which the peaks at 0.92, 1.38, 2.52, 2.83, and 3.27 THz were generated by the weak deformation vibrations of the molecular skeleton of the piperidine group, and the peak at 1.78 THz was formed by the weak deformation vibration of the phenylethyl group (2C-3C); three peaks of methoxyacetylfentanyl were found at 1.11, 1.35, and 2.62 THz. The peaks at 1.11 and 1.35 THz were formed by the out-plane deformation vibrations of 1C-2C and 14C-15C-16C of the acetamide group, respectively. The peak at 2.62 THz was assigned as the deformation vibration of benzene ring on phenylethyl; three peaks of acetylfentanyl were found at 1.07, 1.34, and 2.36 THz, in which the peaks at 1.07 and 2.36 THz were formed by the weak deformation vibrations of piperidine group, and the peak at 1.34 THz was generated by the out-plane deformation vibration between phenyl and ethyl on phenylethyl group (12C-13C); five peaks of furanylfentanyl were found at 0.83, 1.17, 1.70, 2.50, and 2.93 THz. The peak at 0.83 THz was assigned as the in-plane deformation vibration of 11C-10C single bond on piperidine group. The peak at 1.17 THz was assigned as the out-plane deformation vibration of 17C-16C single bond on phenylethyl group. The peaks at 1.70, 2.50, and 2.93 THz were assigned as the deformation vibrations of benzene rings in both benzene and phenethyl, the deformation vibration of benzene ring in phenethyl, and the deformation vibration of piperidinyl molecular skeleton, respectively; three peaks of butyrylfentanyl were found at 0.52, 1.07, and 1.99 THz. The peaks at 0.52 and 1.99 THz were assigned as the deformation vibrations of the molecular skeleton of piperidine groups. The peak at 1.07 THz was assigned as the out-plane deformation vibration of 3C-4C single bond on butyryl group; three peaks of 4-fluoroisobutyrfentanyl were found at 1.19, 1.57, and 2.11 THz. The peak at 1.19 THz was formed by the out-plane deformation vibration of 5C-9N single bond between phenyl and pyridine groups. The peak at 1.57 THz was formed by the in-plane deformation vibration of 7C-6N single bond on piperidine group. The peak at 2.11 THz was formed by the out-plane deformation vibration of 3C-2C-52C on isobutyryl group; four peaks of carfentanil were found at 1.14, 2.00, 2.52 and 3.45 THz. The peaks at 1.14 and 2.52 THz were generated by the deformation vibration of benzene rings in both phenylethyl and phenyl. The peak at 2.00 THz was generated by the deformation vibration of benzene ring in phenyl. The peak at 3.45 THz was generated by the out-plane deformation vibration of 5C-6C single bond on piperidine group; three peaks of valerylfentanyl were found at 0.91, 1.96 and 2.26 THz. The peak at 0.91 THz was assigned as the out- plane deformation vibration of 14C-8N single bond between phenylethyl and piperidine groups. The peak at 1.96 THz was formed by the combined interaction of out-plane deformation vibrations of 9C-10C and 13C-12C on piperidine group. The peak at 2.26 THz was assigned as the out-plane deformation vibration of 10C-11C-12C on piperidine group. 

### 2.4. Identification of Fentanyl Based on Spectral Similarity

To identify the spectral characteristic of fentanyl-related substances that have similar molecular structures, the CS, CCS, DFD, and HD were utilized to calculate the THz spectral similarity between each of the eight fentanyl-related substances, respectively, as seen in Figure 3a–d. The CS curves illustrated in Figure 3a show the relatively low spectral similarity between carfentanil and the other fentanyl-related substances. As shown in Figure 3b, the CCS map shows the similarity values between each two fentanyl analogs, revealing the accuracy of identifying the fentanyl analogs based on their CCS scores. The DFD map (Figure 3c) reveals the spectral distance based on feature points. The smaller the distance, the higher the spectral similarity. The HD curves (Figure 3d) show the relatively higher spectral distance between carfentanil and other fentanyl-related substances. These results indicate that the THz spectral characteristics of carfentanil are more specific than those of other fentanyl analogs. The reason is that carfentanil has a high molecular weight and a unique amide structure as compared to other fentanyl molecules. Therefore, carfentanil can be easily identified among fentanyl analogs by THz spectroscopy. To provide a more intuitive characterizing of spectral similarity, as depicted in Figure 3e, the color bars above the cut-line illustrate the normalized sum of CS and CCS, and the color bars below the cut-line illustrate the normalized sum of DFD and HS. The higher similarity and the corresponding lower distance imply the feasibility of identifying each fentanyl analog. However, acetylfentanyl, butyrylfentanyl, and 4-fluoroisobutyrfentanyl might be misidentified due to their similar spectral characteristics. It is difficult for methods, such as CS, HD, and CCS, to identify spectra with similar feature points through similarity matching. Therefore, a more specific method that matches both the spectral shapes and feature points is needed to identify similar spectra.

### 2.5. Application of Mass Spectral Similarity Mapping to Fentanyl Analogs

The proposed SSM method fuses the spectral shape and peak characteristics for similarity evaluation, providing reliable identification for fentanyl analogs based on stepwise matching from the THz spectral database. This database consisted of the average THz absorption spectra of ten replications of the eight fentanyl analogs. Figure 4 depicts the spectral mapping chains and peak nodes when comparing the target spectrum with that in the database. By calculating the SSM accuracy based on the features of spectral shapes and peaks, the matched spectrum (colored curve) with the highest SSM accuracy was marked to identify the target fentanyl analogy (black curve). Results show that the matched spectrum and the target spectrum has a high degree of similarity, which can successfully identify the category of the target fentanyl with the already known information stored in the spectral database.

The parameters α and β were adjusted to test the performance of the SSM algorithm. The testing results show that with α increasing from 0.1 to 0.6 (interval of 0.1), the top matching list can successfully identify the target fentanyl analog, and the SSM accuracy distribution between the target and that to be matched was stable. Figure 5 plots the spectral similarity mapping to fentanyl analogs by SSR with the parameters α = 0.2 and β = 0.8. The matching list of the target fentanyl analog was sorted in descending order of SSR identification accuracy. The SSR accuracy was calculated to evaluate the probability of identifying the target to the matched fentanyl analog from the database. As can be seen from Figure 5a–h, the target (black curve) was accurately identified with the top hit by the mass spectral similarity mapping. Figure 5i depicts the t-SEN visualization of all the replicated spectra, which further reveals the spectral similarity of fentanyl analogs in mapping space. The main advantage of this algorithm is that it can generate a spectral similarity hit map from searching the THz spectral database. It can accurately identify the fentanyl category of a given spectrum according to the returned top hit results by fusing spectral and feature information, making this method suitable for use as references in forensic laboratories and providing valuable support for combating increasingly rampant abuse of fentanyl-related drugs.

## 3. Materials and Methods

### 3.1. Preparation of Drug Pellets

The real-world fentanyl-containing samples, including fentanyl, methoxyacetylfentanyl, acetylfentanyl, furanylfentanyl, butyrylfentanyl, 4-fluoroisobutyrfentanyl, carfentanil, and valerylfentanyl, were provided by the Chinese Ministry of Public Security. Due to the extremely limited amount of these fentanyl-containing drugs, there was no uniform weighing for each sample. Therefore, the percentage (or purity) of the drug was unknown. These powder samples mixed with polyethylene were prepared as pellets using a compression machine (BJ-15, Tianjin BoJun Technology Co., Ltd., Tianjin, China) with diameters of 13 mm under the constant pressure of 30 MPa for 3 min. These pellets were used for THz spectral measurement without any further preparation.

### 3.2. THz Apparatus and Spectral Measurements

The THz-TDS experimental apparatus (CCT-1800, China Communication Technology Co., Ltd., Shenzhen, China) with its stable transmission-scanning mode, which has been described in detail in our previous works [28], was used to obtain spectra (with spectral resolution of 30 GHz over the range of 0.5–3.5 THz) of the drug pellets. Spectral measurements were performed at ambient temperature of 294 K with dry nitrogen purged into the apparatus to eliminate interference by moisture. The THz spectrum of dry nitrogen was obtained as the reference signal. The average of 100 time-domain scans was obtained as the spectrum of the tested drug sample, and the measurement of each sample was repeated 5 times. Finally, the average value of 5 measurements was taken as the standard spectrum of the test sample to further form the spectral database.

### 3.3. DFT Theoretical Simulations

The Becke three-parameter Lee–Yang–Parr (B3LYP) functional and 6-311G basis set (B3LYP/6-311G) implemented in the Gaussian 2016 software were utilized to optimize the theoretical structures and simulate the theoretical spectra of the 8 fentanyl-containing samples at DFT level. According to the results of DFT theoretical simulation, the THz fingerprint peak properties (e.g., vibrational modes, frequencies, and intensities) of samples were investigated by matching the peaks of the THz experimental spectrum with DFT theoretical spectrum.

### 3.4. Spectral Similarity Calculations

To facilitate the spectral searching and matching process, the sample spectrum and its standard spectrum were selected for spectral similarity calculation. The methods of CS and CCS were used to evaluate the THz spectral similarity between fentanyl-related substances, as well as the methods of HD and DFD were used to calculate the distance between two spectra (The smaller the distance between two vectors, the higher the similarity). CS is a measure of similarity between two vectors of an inner product space that measures the cosine of the angle between them (The cosine of 0° is 1, and it is less than 1 for any other angle) [29]. CCS is a function that calculates cosine–cross similarity between the time series of threshold crossings with time lags. It also calculates the dominant lag and maximum value [30]. HD is a technique used to determine differences between spectral shapes as well as the degree of difference between two shapes [31]. DFD calculates the discrete Fréchet distance between curves P and Q. P and Q are two sets of points that define polygonal curves with rows of vertices (data points) and columns of dimensionality. Points along the curve are taken in the order that they appear in P and Q [32]. These methods are efficient in evaluating spectral similarity and easy to be realized. However, they cannot provide quantitative evaluation results by one single comparison of the two spectra. Even with an exhaustive pairwise comparison of the spectra, the evaluation results are still far from the actual multi-source spectral similarity, which does not fundamentally solve the issue of multi-source spectral similarity evaluation. Herein, we present a SSM method that fuses weighted CS (Equation (1)) and DFD (Equation (2)) to directly evaluate the spectral matching accuracy from two scales of spectral shape and spectral characteristics. The reason for fusing CS and DFD is that they are efficient methods to evaluate spectral similarity (i.e., between two curves) and characteristic distance (i.e., between two sets of feature points on curves), respectively. In addition, they are easy to be realized without any parameters. The spectral matching accuracy can be calculated as follows:(1)$${\mathrm{CS}}_{i}=\cos\left(\frac{180}{\pi }\times \arccos\left[\frac{S_{t}S_{i}{}^{T}}{\left|S_{t}\right|\left|S_{i}\right|}\right]\right)$$
(2)$${\mathrm{DFD}}_{i}=\min\{\underset{j=1,2,\cdots ,m}{\max}d(\upsilon s_{i}^{j},\upsilon s_{t}^{j})\}$$
(3)$$Acc=\alpha \cdot {\mathrm{CS}}_{i}+\beta \cdot \left(1-\frac{{\mathrm{DFD}}_{i}}{\sum_{i=1}^{n}{\mathrm{DFD}}_{i}}\right)$$
where *n* (*I* = 1, 2, …, *n*) is the number of spectrum to be matched with a target. *CS_i_* and DFD*_i_* are the CS and DFD between the *i*-th spectrum (*S_i_*) and the target spectrum (*S_t_*), respectively. *m* (*j* =1, 2, …, *m*) is the number of features on the spectrum, $d(\upsilon s_{i}^{j},\upsilon s_{t}^{j})$ is the Fréchet distance between features of $\upsilon s_{i}^{j}$ and $\upsilon s_{t}^{j}$ (on the spectrum of *S_i_* and *S_t_*, respectively). *α* and *β* (*α* + *β* = 1) are the weights of CS and DFD, respectively. *Acc* is the accuracy of specifying the *i*-th spectrum as the target spectrum.

## 4. Conclusions

This work demonstrates the identification of fentanyl-related drugs with similar molecular characteristics by THz spectral analysis. The formation mechanism of the THz peaks of these drugs was investigated by Gaussian-accelerated molecular dynamics simulations, further explaining how these fentanyl molecules interact with the THz radiation. It has been shown that it was difficult to identify fentanyl analogs that were being created with slight modifications to the molecular functional groups by the direct comparison of THz spectral peaks or spectral curves due to their similar spectral features. Here, we presented a fentanyl analog identification method based on THz spectral similarity mapping that incorporated spectral curves and peak nodes. This method allowed the accurate identification of fentanyl analogs based on the top hit list returned by the stepwise searching and matching the database. Furthermore, as most traditional machine learning methods required a large dataset for training and testing models, our method focused on revealing spectral characteristics without requiring large datasets. The target fentanyl analog could be accurately identified with the top hit result. This work demonstrated the efficacy of a functional data analysis to identify molecules containing similar functional groups from their THz absorption spectra, providing valuable support for combating increasingly rampant fentanyl abuse.

## Figures and Tables

**Figure 1 ijms-23-10321-f001:**
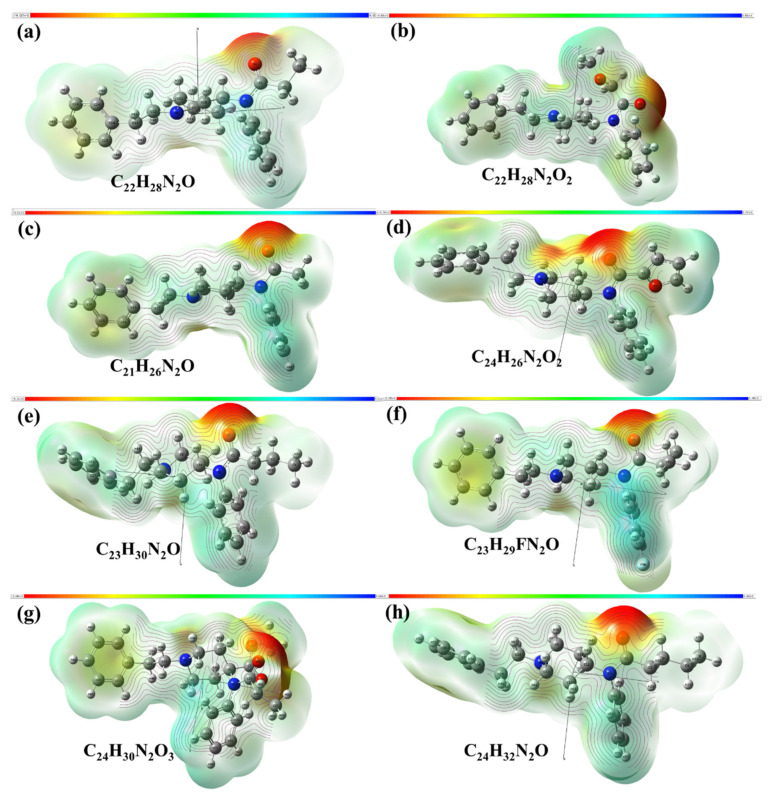
Geometric configuration of fentanyl molecules of (**a**) fentanyl, (**b**) methoxyacetylfentanyl, (**c**) acetylfentanyl, (**d**) furanylfentanyl, (**e**) butyrylfentanyl, (**f**) 4-fluoroisobutyrfentanyl, (**g**) carfentanil, and (**h**) valerylfentanyl.

**Figure 2 ijms-23-10321-f002:**
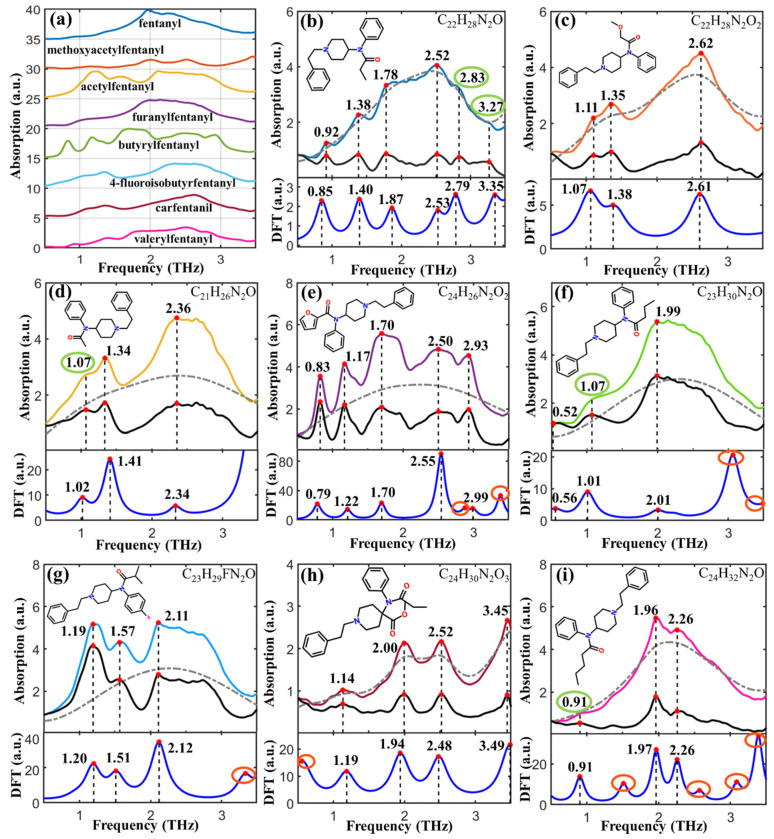
THz spectra of (**a**) eight fentanyl-related substances and spectral peak matching of (**b**) fentanyl, (**c**) methoxyacetylfentanyl, (**d**) acetylfentanyl, (**e**) furanylfentanyl, (**f**) butyrylfentanyl, (**g**) 4-fluoroisobutyrfentanyl, (**h**) carfentanil, and (**i**) valerylfentanyl.

**Figure 3 ijms-23-10321-f003:**
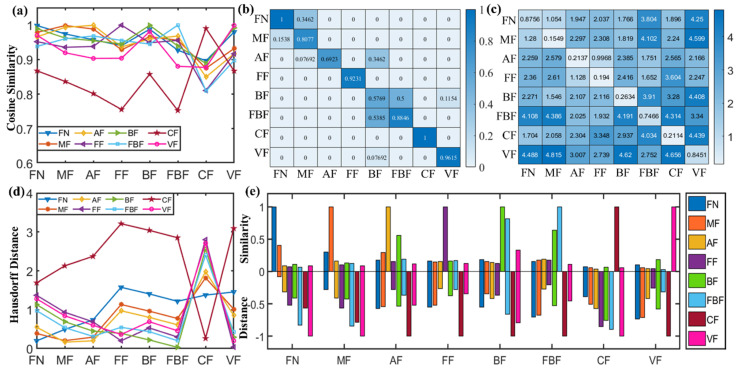
THz spectral similarity calculation for the eight fentanyl analogs based on (**a**) CS, (**b**) CCS, (**c**) DFD, (**d**) HD, and (**e**) the statistical histogram based on these four methods. Note: fentanyl, methoxyacetylfentanyl, acetylfentanyl, furanylfentanyl, butyrylfentanyl, 4-fluoroisobutyrfentanyl, carfentanil, and valerylfentanyl are abbreviated as FN, MF, AF, FF, BF, FBF, CF, and VF, respectively, in this figure.

**Figure 4 ijms-23-10321-f004:**
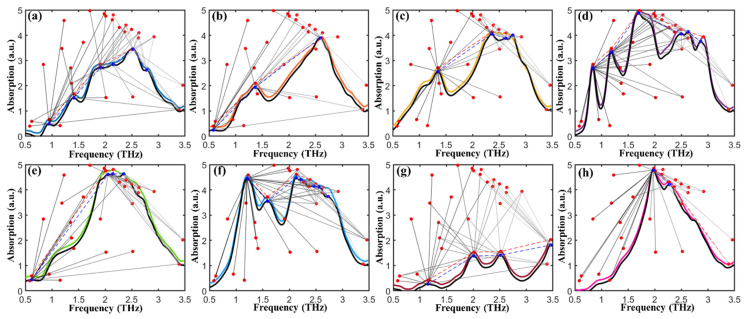
Feature points searching and matching for (**a**) fentanyl, (**b**) methoxyacetylfentanyl, (**c**) acetylfentanyl, (**d**) furanylfentanyl, (**e**) butyrylfentanyl, (**f**) 4-fluoroisobutyrfentanyl, (**g**) carfentanil, and (**h**) valerylfentanyl.

**Figure 5 ijms-23-10321-f005:**
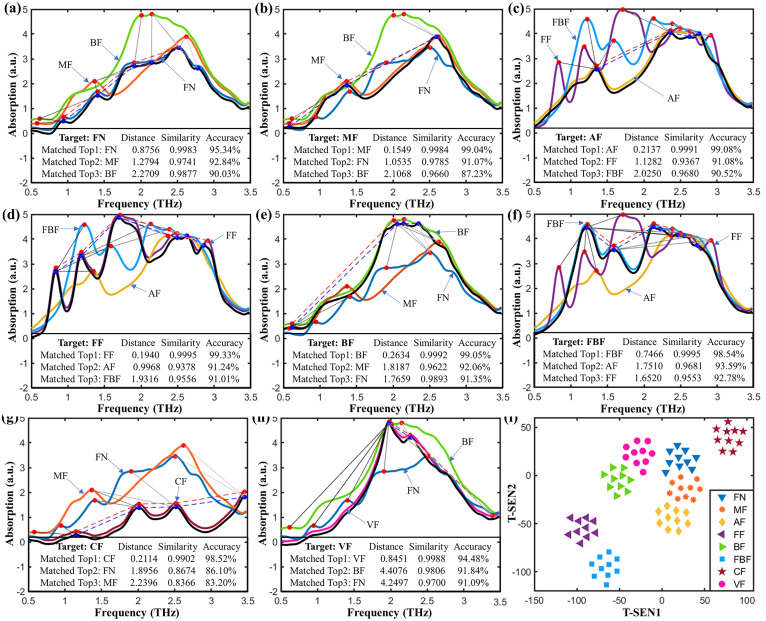
Mass spectral similarity mapping to fentanyl analogs of (**a**) fentanyl, (**b**) methoxyacetylfentanyl, (**c**) acetylfentanyl, (**d**) furanylfentanyl, (**e**) butyrylfentanyl, (**f**) 4-fluoroisobutyrfentanyl, (**g**) carfentanil, (**h**) valerylfentanyl, and (**i**) t-SEN visualization. Note: fentanyl, methoxyacetylfentanyl, acetylfentanyl, furanylfentanyl, butyrylfentanyl, 4-fluoroisobutyrfentanyl, carfentanil, and valerylfentanyl are abbreviated as FN, MF, AF, FF, BF, FBF, CF, and VF, respectively, in this figure.

**Table 1 ijms-23-10321-t001:** Assignment of the absorption peaks.

DFT Peaks (THz)	THz Peaks (THz)	Shift (THz)	Vibration Modes
Fentanyl			
0.85 (w)	0.92	−0.07	δring
1.40 (w)	1.38	0.02	δring
1.87 (w)	1.78	0.09	δ(C-C)
2.53 (w)	2.52	0.01	δring
2.79 (w)	2.83	−0.04	δring
3.35 (w)	3.27	0.08	δring
Methoxyacetylfentanyl			
1.07 (w)	1.11	−0.04	δ(C-C) oop
1.38 (w)	1.35	0.03	δ(C-C-C) oop
2.61 (w)	2.62	−0.01	δring
Acetylfentanyl			
1.02 (w)	1.07	−0.05	δring
1.41 (m)	1.34	0.07	δ(C-C)opp
2.34 (w)	2.36	−0.02	δring
Furanylfentanyl			
0.79 (s)	0.83	−0.04	δ(C-C)ip
1.22 (m)	1.17	0.05	δ(C-C)opp
1.70 (m)	1.70	0	δring
2.55 (vs)	2.50	0.05	δring
2.99 (w)	2.93	0.06	δring
Butyrylfentanyl			
0.56 (w)	0.52	0.01	δring
1.01 (vs)	1.07	−0.06	δ(C-C)opp
2.01 (w)	1.99	0.02	δring
4-Fluoroisobutyrfentanyl			
1.20 (m)	1.19	0.01	δ(C-C)opp
1.51 (m)	1.57	−0.06	δ(C-N)ip
2.12 (s)	2.11	0.01	δ(C-C)opp
Carfentanil			
1.19 (w)	1.14	0.05	δring
1.94 (m)	2.00	−0.06	δring
2.48 (m)	2.52	−0.04	δring
3.49 (m)	3.45	0.04	δ(C-C)opp
Valerylfentanyl			
0.91 (m)	0.91	0	δ(C-N)opp
1.97 (m)	1.96	0.01	δ(C-C)opp
2.26 (m)	2.26	0	δ(C-C-C)opp

oop: out-plane bending; ip: in-plane bending; δ: deformation vibration; vs: very strong; s: strong; m: medium; w: weak; vw: very weak.

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
