# Peer review of "High-Precision Automatic Identification of Fentanyl-Related Drugs by Terahertz Spectroscopy with Molecular Dynamics Simulation and Spectral Similarity Mapping"

_ijms, 2022, doi:10.3390/ijms231810321_

Round 1

Reviewer 1 Report

The topic of this article is interesting, the authors presenting a rapid and sensitive method for the identification of fentanyl-related drugs using terahertz spectroscopy with molecular dynamics simulation and spectral similarity mapping.

After reading the manuscript, the following doubts and suggestions have arisen.

In the abstract the background, purpose, methods, results, and conclusion should be proportionally and clearly presented. A graphical abstract would be appreciated.

The introduction and the discussion sections should be more complete, providing supplementary background about the physical, chemical and pharmacokinetics differences between the fentanyl-related compounds investigated, as well as, about the communicated data on the other methods for their detection (see:

  • Zhang W et al. Investigation of THz absorptive signatures in opioids. Appl. Sci. 2022, 12, 61.
  • Ye J et al. Surface-enhanced shifted excitation Raman difference spectroscopy for trace detection of fentanyl in beverages. Appl. Opt. 2021; 60, 2354-2361.
  • Forbes TP et al. Discriminative potential of ion mobility spectrometry for the detection of fentanyl and fentanyl analogues relative to confounding environmental interferents. Analyst. 2019; 144(21):6391-6403.

·         Haddad A et al. Detection and Quantitation of Trace Fentanyl in Heroin by Surface-Enhanced Raman Spectroscopy. Anal. Chem. 2018, 90, 21, 12678-12685.

The results obtained should be compared with those achieved by other researchers and discussions should be significantly detailed.

In discussion section, the authors need to develop argumentation in depth based on the current understanding and the findings of this study, presenting the potential, the weakness and limitation, and future research direction, among others. Authors should try to explain the theoretical implication as well as the translational application of their research.

Some other aspects were found in this manuscript:

- all abbreviations should be expanded in the first appearance. The explanation of the abbreviation should be used only once in the text and should not be repeated, in order to decongest the text and facilitate the understanding of the information transmitted.

- spelling check of the text is mandatory.

- the authors should upgrade the references; doi numbers should be provided to each reference.

- English including grammar, style and syntax, should be improved through the professional help from English Editing Company for Scientific Writings.

Author Response

>1) In the abstract the background, purpose, methods, results, and conclusion should be proportionally and clearly presented. A graphical abstract would be appreciated.

Response: Weve added the graphical abstract to show the main work of this manuscript in re-submission.

>2) The introduction and the discussion sections should be more complete, providing supplementary background about the physical, chemical and pharmacokinetics differences between the fentanyl-related compounds investigated, as well as, about the communicated data on the other methods for their detection (see:Zhang W et al. Investigation of THz absorptive signatures in opioids. Appl. Sci. 2022, 12, 61.Ye J et al. Surface-enhanced shifted excitation Raman difference spectroscopy for trace detection of fentanyl in beverages. Appl. Opt. 2021; 60, 2354-2361.Forbes TP et al. Discriminative potential of ion mobility spectrometry for the detection of fentanyl and fentanyl analogues relative to confounding environmental interferents. Analyst. 2019; 144(21):6391-6403. Haddad A et al. Detection and Quantitation of Trace Fentanyl in Heroin by Surface-Enhanced Raman Spectroscopy. Anal. Chem. 2018, 90, 21, 12678-12685.) The results obtained should be compared with those achieved by other researchers and discussions should be significantly detailed. In discussion section, the authors need to develop argumentation in depth based on the current understanding and the findings of this study, presenting the potential, the weakness and limitation, and future research direction, among others. Authors should try to explain the theoretical implication as well as the translational application of their research.

Response: The main differences between the fentanyl-related substances lied in the composition of their elements. As mentioned in Line 35-39, the pharmacological effects could be retained or enhanced by modifying the functional group on fentanyl. The main structural transformations include replacement of the phenethyl group with various other groups (except hydrogen); introduction of substituents, such as methyl groups and methoxycarbonyl groups, at different positions on the piperidine ring; replacement of the propionyl group in the 4-anilido fragment with various other acyl groups; and replacement of the whole benzene ring attached to the nitrogen atom with substituted or unsubstituted monocyclic aromatic heterocycles. In this way, different fentanyl-related substances could be synthesized, they share different number of constituent elements (C, H, N, O) but similar molecular structures and similar physical, chemical and pharmacokinetics properties.

After carefully reading the articles mentioned above, with respectfully disagree on the comment that we should compare our results with those achieved by other researchers. Because in our work, we focused on analyzing the THz fingerprints of the studied substances and identifying these substances with similar spectral characteristics efficiently, rapidly, and accurately by our proposed method. Hence, it is difficult to compare the papers that with different research objectives (trace detection VS automatic recognition) and different spectroscopy techniques (ion mobility spectrometry, Surface-Enhanced Raman Spectroscop VS Terahertz).

In discussion section, we added the potential future applications of our method as It can accurately identify the fentanyl category of a given spectrum according to the returned top hit results by fusing spectral and feature information, making this method suitable for use as references in forensic laboratories and providing valuable support for combating increasingly rampant abuse of fentanyl-related drugs in Line 199-202.

>3) Some other aspects were found in this manuscript:

-a) all abbreviations should be expanded in the first appearance. The explanation of the abbreviation should be used only once in the text and should not be repeated, in order to decongest the text and facilitate the understanding of the information transmitted.

-b) spelling check of the text is mandatory.

-c) the authors should upgrade the references; doi numbers should be provided to each reference.

-d) English including grammar, style and syntax, should be improved through the professional help from English Editing Company for Scientific Writings.

Response:  We apologize for any mistake in the manuscript. To tackle the errors, this manuscript had been professionally edited, checked, and polished by two native English speakers with Ph.D. degrees from International Science Editing Service (http://www.internationalscienceediting.com) before we resubmit it. Also, weve checked this manuscript carefully for several rounds to avoid any mistake. Now, we can guarantee that there is no typo in this resubmission. We believe the manuscript has been much improved, and also well organized. In addition, the DOI numbers were added in the reference section.

Reviewer 2 Report

This manuscript is devoted to studying fentanyl and its derivatives with using THz time-domain spectroscopy (THz-TDS) technique coupled with density functional theory (DFT) and spectral similarity mapping(SSM). The approach to identification of fentanyl analogs based on the top hit list returned by the stepwise searching and matching the database is presented in this manuscript. It is shown that revealing the spectral characteristics without requiring large datasets allow to distinguish fentanyl and its analogs. These results can be used in future as an artificial-intelligence-based tool for high-precision fentanyl analysis and can be interesting for technical and scientific groups in area of spectroscopy, medicine, drug control authorities etc.

There are some points to correct or to make the information more clear:

1)      It is necessary to explain of colors in the Figure 1: as colors of molecular electrostatic potential (there are color scales on top of every part of this Figure, but if any marks are presented in this scale, they are very small even at zoom), as colors for atoms in molecules. Although all atoms have the marks but these marks are very small and are hard to distinguish because of bad resolution even at zoom. Also it will be convenient if the structural formulas of the fentanyl and fentanyl’s derivatives will be added for every substances because these structural formulas without spatial structure used further (e.g. figure 2) and reader needs to guess e.g. the benzene rings turned sideways to the viewer as for acetylfentanyl. 

2)      Some notes for Figure 2 and its description in text. Authors write: (130th-132nd  lines) “However, several 130 spectral characteristics, e.g., 2.83 THz and 3.27 THz of fentanyl [Fig. 2(a)], 1.07 THz of both acetylfentanyl [Fig. 2(c)] and butyrylfentanyl [Fig. 2(e)], and 0.91 THz of valerylfentanyl 132 [Fig. 2(h)]…” and “THz spectral baseline correction, peak finding and peak matching with DFT of (a) fentanyl, 149 (b) methoxyacetylfentanyl, (c) acetylfentanyl, (d) furanylfentanyl, (e) butyrylfentanyl, (f) 4-150 fluoroisobutyrfentanyl, (g) carfentanil and (h) valerylfentanyl” in Figure’s caption. But the first part of Figure 2 marked “(a)” is collection of spectra of fentanyl and 7 its derivatives and the spectrum of fentanyl is presented in Figure2(b). All other spectra must be also moved, respectively, in description and in Figure caption. Besides it is necessary to explain the Y-axis in Figure 2a (that the 0 level for all substances excepting valerylfentanyl has shift), because the absorption of every substance is limited by values of 0-6 a.u., but the absorption values presented in figure 2a with collected spectra is e.g. from 35 to 40 a.u. for fentanyl etc. Besides the X-axis captions in Figure 2(d-f) are cut off.  In addition it is necessary to explain that the peaks in green rings in the some parts of Figure 2 (b,d,f,i) are apparently the “several spectral characteristics, e.g., 2.83 THz and 3.27 THz of fentanyl [Fig. 2(a)], 1.07 THz of both  acetylfentanyl [Fig. 2(c)] and butyrylfentanyl [Fig. 2(e)], and 0.91 THz of valerylfentanyl 132 [Fig. 2(h)]” which ”were difficult to be determined” (130th – 133rd lines). Besides there are not any explanations about red rings at DFT curves  in the Figure 2 (e-i). What does dash-dotted line mean?

3)      Authors writes: “As shown from Fig.3(b), the CCS map shows the similarity values between each two fentanyl analogs, revealing 201 the accuracy of identifying the fentanyl analogs based on their CCS scores”. It is not clear why the diagonal values  (Fig.3b) sometimes far from 1. Is there bad similarity for data  for acetylfentanyl with itself or butyrylfentanyl with itself? Besides, it is not clear, why the data for CCS similarity for two couples of substances are different about two times (MF(FN)=0.1538, FN(MF)=0.3462). The same situation with slightly smaller difference is for valerylfentanyl and butyrylfentanyl.

4)      Some notes for Figure 5. All parts of Figure have 4 curves: curve for specific substance, its target and curves for two another substances.  But there are 8 substances studied here. Are other substances considered? Besides the X-axis captions in Figure 5(d-f) are cut off. What do red and blue dashed lines here and in Figure 4 mean?

This manuscript is written sufficiently clear and describes with details the results of development of the method of processing and mapping of time-domain THz spectral data for fentanyl and fentanyl derivatives.

The manuscript can be published after minor revisions.

Author Response

>1) It is necessary to explain of colors in the Figure 1: as colors of molecular electrostatic potential (there are color scales on top of every part of this Figure, but if any marks are presented in this scale, they are very small even at zoom), as colors for atoms in molecules. Although all atoms have the marks but these marks are very small and are hard to distinguish because of bad resolution even at zoom. Also it will be convenient if the structural formulas of the fentanyl and fentanyl’s derivatives will be added for every substances because these structural formulas without spatial structure used further (e.g. figure 2) and reader needs to guess e.g. the benzene rings turned sideways to the viewer as for acetylfentanyl.

Response: Thanks for your comment. We’d like to explain that it is easy to identify atoms (C, H, N, O) in this current scale according to their color, as well as with the assistance of structural formulas shown below each molecular structure in Figure 1. Based on this information, we can easily know that all of the 8 fentanyl-related substances contain 2 N-atoms (in blue), 1~3 O-atoms (in red), 26~32 H-atoms (in light grey), and 21~24 C-atoms (in dark grey). So we marked this key information in Line 92-93. In Figure 1, the electrostatic potential value of the red region is negative, which indicates that this region is easier to give electrons, or more nucleophilic than other regions; The positive electrostatic potential in the blue region indicates that this region is easier to obtain electrons and is more electrophilic than other regions. In addition, we added the corresponding structural formulas of each substances in Figure 2.

>2) Some notes for Figure 2 and its description in text. Authors write: (130th-132nd  lines) “However, several 130 spectral characteristics, e.g., 2.83 THz and 3.27 THz of fentanyl [Fig. 2(a)], 1.07 THz of both acetylfentanyl [Fig. 2(c)] and butyrylfentanyl [Fig. 2(e)], and 0.91 THz of valerylfentanyl 132 [Fig. 2(h)]…” and “THz spectral baseline correction, peak finding and peak matching with DFT of (a) fentanyl, 149 (b) methoxyacetylfentanyl, (c) acetylfentanyl, (d) furanylfentanyl, (e) butyrylfentanyl, (f) 4-150 fluoroisobutyrfentanyl, (g) carfentanil and (h) valerylfentanyl” in Figure’s caption. But the first part of Figure 2 marked “(a)” is collection of spectra of fentanyl and 7 its derivatives and the spectrum of fentanyl is presented in Figure2(b). All other spectra must be also moved, respectively, in description and in Figure caption. Besides it is necessary to explain the Y-axis in Figure 2a (that the 0 level for all substances excepting valerylfentanyl has shift), because the absorption of every substance is limited by values of 0-6 a.u., but the absorption values presented in figure 2a with collected spectra is e.g. from 35 to 40 a.u. for fentanyl etc. Besides the X-axis captions in Figure 2(d-f) are cut off.  In addition it is necessary to explain that the peaks in green rings in the some parts of Figure 2 (b,d,f,i) are apparently the “several spectral characteristics, e.g., 2.83 THz and 3.27 THz of fentanyl [Fig. 2(a)], 1.07 THz of both  acetylfentanyl [Fig. 2(c)] and butyrylfentanyl [Fig. 2(e)], and 0.91 THz of valerylfentanyl 132 [Fig. 2(h)]” which ”were difficult to be determined” (130th – 133rd lines). Besides there are not any explanations about red rings at DFT curves in the Figure 2 (e-i). What does dash-dotted line mean?

Response: Thank you for pointing this out. We apologize for these mistakes related to Figure 2 and its descriptions. Weve renewed this whole paragraph carefully in the revised manuscript. Wed like to explain that the spectra of all these eight fentanyl-related substances were shown hierarchically (Fig 2a), and the Y-axes (THz absorptance) were relatively between 0~5 a.u.

The X-axis (0.5 ~ 3.5 THz) captions (Frequency (THz)) were already shown in all subfigures of Fig 2. Although the subfigures of Figure 2(d-f) were divided into upper and lower parts, they shared the same X-axis and caption.

The peaks in green rings were those at 2.83 THz and 3.27 THz of fentanyl [Fig. 2(b)], 1.07 THz of both acetylfentanyl [Fig. 2(d)] and butyrylfentanyl [Fig. 2(f)], and 0.91 THz of valerylfentanyl [Fig. 2(i)], which were difficult to be determined whether these are the fingerprint peaks or not due to low resolution. The peaks in red rings were those in DFT spectra, which were unpaired with those in THz spectra. We mentioned these key information in the revised manuscript in Line 104-105, and Line 111, respectively.

>3) Authors writes: “As shown from Fig.3(b), the CCS map shows the similarity values between each two fentanyl analogs, revealing 201 the accuracy of identifying the fentanyl analogs based on their CCS scores”. It is not clear why the diagonal values  (Fig.3b) sometimes far from 1. Is there bad similarity for data for acetylfentanyl with itself or butyrylfentanyl with itself? Besides, it is not clear, why the data for CCS similarity for two couples of substances are different about two times (MF(FN)=0.1538, FN(MF)=0.3462). The same situation with slightly smaller difference is for valerylfentanyl and butyrylfentanyl.

Response: Firstly, we apologize for the unclear explanation of results in Figure 3(b). Wed like to explain that the diagonal values were calculated between the sample spectrum (the first measurement) and its standard spectrum (the average of 5 measurement to further form the spectral database). These two spectra of the same sample (on-diagonal elements) were not exactly the same. Therefore, there would be results far from 1 or different CCS scores for two pairs of substances (off-diagonal elements). To standard the content clearly, we added some key information in the revised manuscript.

Line 222-225: The average of 100 time-domain scans was obtained as the spectrum of the tested drug sample, and the measurement of each sample was repeated 5 times. Finally, the average value of 5 measurements was taken as the standard spectrum of the test sample to further form the spectral database. (Section 3.2 THz apparatus and spectral measurements)

Line 233-234: To facilitate the spectral searching and matching process, the sample spectrum and its standard spectrum were selected for spectral similarity calculation. (Section 3.4 Spectral similarity calculations)

>4) Some notes for Figure 5. All parts of Figure have 4 curves: curve for specific substance, its target and curves for two another substances. But there are 8 substances studied here. Are other substances considered? Besides the X-axis captions in Figure 5(d-f) are cut off. What do red and blue dashed lines here and in Figure 4 mean?

Response: Thanks for your comment. Actually, we did consider all of these 8 substances. For each of the specific substance, the top three matched analogs were retrieved based on our proposed method. As shown in Figure 5, the matched top 1 (standard spectrum from the database) is the target (sample spectrum), which means our method is highly accurate in identifying the category of fentanyl-related substances. The matched top 2 and top 3 are two another substances, which have similar spectral characteristics with the target. With respectfully remind that the X-axis captions (Frequency THz) in Figure 5(d-f) have already been presented. The red and blue dashed lines in Figure 4 are the links between feature points.

Round 2

Reviewer 1 Report

The authors mostly responded to the comments and suggestions and the manuscript was revised accordingly. I consider it could be accepted for publication in this journal.